# Long COVID cognitive sequelae 6 months postinfection and beyond: a scoping review protocol

Sara Monteiro [1], Coralie Dessenne [2], Magali Perquin [1]

¹Department of Precision Health, Luxembourg Institute of Health, Strassen, Luxembourg
²Science Office, Luxembourg Institute of Health, Strassen, Luxembourg

**Correspondence to**
Ms Sara Monteiro;
sara.monteiro@lih.lu

## ABSTRACT

**Introduction** The novel and expanding field of long COVID research has undergone diverse methodological approaches in recent years. This protocol lays out the methodological approach, which aims at identifying nuances in current research. It underscores the necessity for a more precise understanding of prolonged cognitive sequelae and their relation to initial disease severity. The findings will add valuable insights for the development of targeted rehabilitation, healthcare interventions and thereby aid patients, clinicians, policymakers and researchers. Our upcoming research is introduced here.

**Methods and analysis** To map current research in the field, a scoping review will be conducted and documented in accordance with the Preferred Reporting Items for Systematic Reviews and Meta-Analyses for Scoping Review Extension standards. A systematic search of scientific databases (PubMed, EMBASE), presented 1409 eligible results, published up to 21 December 2023. After removal of duplicates, 925 articles were extracted for screening. Two independent reviewers will screen for titles, abstracts and full texts, to extract data, which will then be organised using charting software. Data for various variables, that is, journal info, studied population demographics, study design, long COVID related data, cognitive outcomes and neuropsychological tests will be gathered. Descriptive analyses, evidence gap maps, heat map quantifications and narrative synthesis will be conducted for reporting of results.

This scoping review has been registered with the Open Science Framework (https://doi.org/10.17605/OSF.IO/JHFX6).

**Ethics and dissemination** Ethical approval is not required, as the study does not involve human participants. The findings will be disseminated through a publication in a scientific journal and within the professional network.

## STRENGTHS AND LIMITATIONS OF THIS STUDY

⇒ Cognitive symptoms will be characterised in detail, highlighting domains and particular subskills for in depth understanding of long COVID pathology.

⇒ The multifaceted approach used (in-depth psychometrics, acute and long-term COVID-19 symptomatology, as well as comorbid factors), enables a nuanced analysis and increases accuracy.

⇒ A multidisciplinary team covering information science, neuropsychology and epidemiology add to the depth of interpretability and robustness of the review, by establishing a comprehensive search strategy and evaluation.

⇒ Evidence maps will be used to address present heterogeneity in the field and thereby allowing for improved scientific understanding and enhanced clinical interventions.

## BACKGROUND

According to the WHO, 10%–20% of individuals who contract SARS-CoV-2 infection may go on to experience chronic symptoms,[1] that might persist from weeks and up to years.[2] The overarching label for persistent symptoms is 'long COVID'[2,3] alternatively, 'postacute sequelae of SARS-CoV-2 infection'[4] 'post-COVID-19 syndrome (PCS)'[5,6] and 'post-COVID-19 condition'[7] have been proposed.[2,3] Debates exploring consensus for a unified syndrome definition, labels and symptoms and their temporal evolution are currently ongoing. Establishing a unified model is challenging, particularly due to the varying severity, duration and overall heterogeneity of symptoms, the novelty of the field and a lack of unified assessment approaches used in studies,[2,8] consequently also affecting the availability of healthcare management.[3]

Attempts have been made to map out the characteristics of PCS.[9] Three core categories seem to dominate the clinical picture, namely cognitive, psychiatric and systemic issues.[9] Consensus seems to exist for frequent symptoms of the systemic core, which includes fatigue, headaches, anosmia/dysgeusia, myalgia and respiratory difficulties, as well as the psychiatric core, mainly involving anxiety and depression.[9,10] Memory, executive functioning and attentional deficits have been reported to be at the core of cognitive complaints, as well as cognitive slowing, commonly termed 'brain fog'.[9]

By dividing the condition into temporal stages, a number of models have been put forth during the last three years to explain and categorise the status of chronic symptoms.[2,4]

The majority of reviews and studies up to now focus on the early and acute phases following infection.[6–10] Recent scoping reviews on COVID-19 reported cognitive symptoms and tests being used for screening up to six months maximum after the acute phase,[9] as well as psychiatric conditions prior to and immediately after infection.[11 12] Conversely, there has been a less comprehensive examination, especially regarding the neurocognitive symptom core, for long COVID symptoms beyond these immediate stages.[8 9 13] However, a comparison between immediate and long-term effects has been reviewed, although on specific populations, such as patients with substance use disorders.[14] Thus, although existing, to our knowledge, no study has addressed the specific and recurrent cognitive symptoms beyond 6 months after infection.

It is thus important to improve understanding of the current gap in the field relating to cognitive long-term sequelae, by identifying which symptoms are most commonly present in patients often referred to as 'long haulers.' Because long COVID has recently been found to be related with grey matter structural changes [13] and permanent alteration of brain connectivity, even eleven months after infection,[15] it is essential to map the exact cognitive decline associated with the persistent manifestation.[16] Attention, executive and memory components seem to lie at the core of the difficulties,[9] however there is currently no single, cohesive concept about their persistency, which exact deficits are present to what extent and with regards to which initial severity, or on how to measure long COVID associated cognition.[8 9] Additionally, it is necessary to provide clarification regarding the heterogeneous use of study designs, assessment techniques, confounding caused by coexisting conditions that were not tested for, and a lack of comparison to earlier tests or controls.[9] This approach will help with the optimisation of existing medical diagnostic techniques, subsequent treatment plans and rehabilitative strategies and add to our understanding of the exact components of long COVID associated cognitive impairment.

## OBJECTIVES

The aim of this scoping review is to map out the persistent cognitive sequelae associated with long COVID symptomatology in adults, 6 months or more after infection.

► Characterise patterns of long COVID cognitive sequelae over time, specifically chronic symptoms present after at least 6 months or beyond.
► Investigate whether the initial disease severity (asymptomatic, mild, moderate, severe) influences the cognitive outcomes.
► Report on neuropsychological tests for cognitive assessment that were used to assess the cognitive changes, to understand the conceptual boundaries of current findings.
► Elucidate current gaps in the literature.

## METHODS AND ANALYSIS

### Protocol and registration

The protocol is registered under the Open Science Framework with OSF (https://doi.org/10.17605/OSF.IO/JHFX6).

### Design

To guide the overall conception and reasoning, the Arksey and O'Malley methodological framework [17] will be applied to the design and decision-making process of the systematic search. Additionally, the methodological concept proposed by the National Institute for Health Research Service Delivery and Organisation Research and Development Programme [18] will be used, with a main focus on (1) clarifying conceptual working definitions and to determine current boundaries of the field and concept, (2) identifying existing knowledge and present gaps and (3) mapping existing evidence.[19] Reporting the work will be aligned with the Preferred Reporting Items for Systematic Reviews and Meta-Analyses (PRISMA) guidelines for systematic reviews and the PRISMA Extension for Scoping Reviews (PRISMA-ScR) [20] and will follow the rationale of the PRISMA 2020 statement update,[21] in the context of a scoping study.

To the best of our knowledge, after a preliminary search in November 2023 of PubMed, Google Scholar, OSF, PROSPERO and the Cochrane Database of Systematic Reviews, no previous or upcoming reviews or protocols with a comparable design have been identified.

### Patient and public involvement

No patients involved.

### Eligibility criteria

As shown in table 1, the PICO framework [22] served as a framework for determining the eligibility requirements.

### Information resources

Articles were extracted from the following databases:
► PubMed.
► EMBASE.

### Search strategy

The search strategy was constructed together with a scientific librarian, to conduct a wide-ranging search within the selected databases, aiming at identification of an unbiased body of records. As outlined in table 2, search terms from three themes ('long COVID', 'cognition' and 'assessment') will be conceptually arranged to be placed in comprehensive search strings for selection of relevant articles. Within a theme, search terms are arranged in conceptual clusters, which will be combined by syntax techniques such as Boolean operators ('OR', 'AND'), search tags (MeSH, title only) and filters ('humans', 'English language'). Additionally, word truncation (*) will be used to capture alternate word-endings. For example, the search term "COVID*" will capture instances of the word "COVID", and also include other labels or terms containing "COVID" within them (eg, "long-COVID",

**Table 1** Eligibility criteria based on the PICO framework

| Element | Inclusion criteria | Exclusion criteria |
|---|---|---|
| Participants | Human participants<br>Articles involving adults ≥18 years old<br>Contains reliable information on previous COVID-19 infection<br>Chronic, recurrent or onset of cognitive symptomatology ≥6 months after infection | Animal studies<br>Articles focusing on patients <18 years old |
| Intervention or exposure of interest | No intervention reviewed; exposure is (1) SARS-CoV-2 initial infection, (2) cognitive testing after 6 months<br>Chronic, recurrent or onset of cognitive symptomatology ≥6 months after infection<br>Preferred inclusion of studies reporting exclusion of people with preinfection cognitive symptoms or disorders<br>Included studies report on tools used for identification of cognitive symptoms<br>At least one objective, validated and quantifiable cognitive assessment tool used | Articles only reporting outcomes for symptom duration <6 months<br>Articles which do not include cognitive symptoms |
| Comparator | The studies should include cognition outcomes long time after COVID-19<br>Different cognitive symptoms of interest<br>Different types of initial disease severity (which could have required different types of medical care, eg, hospitalisation)<br>Different assessment modalities for cognitive performance | No information available about long COVID after 6 months or beyond<br>No initial infection to SARS-CoV-2 |
| Outcome | Any neurocognitive outcome related to long COVID characteristics<br>'Long COVID' is not restricted to a specific definition or symptomatology, but exploration thereof remains open and will be explored<br>Alternative labels will be accepted<br>Any objective cognitive assessment modality will be included | Self-report of cognitive ability without objective measurement tool |
| Study design | Any type of study that assists in scoping the field, independent of the type, will be included (preference for longitudinal, transversal, with a cohort or a case–control design, review, meta-analysis) | Studies with n<10 (eg, case reports), qualitative studies |
| Article type | Peer-reviewed articles<br>Available in full text | Grey literature, protocols, editorials, dissertations, book chapters, conference abstracts |
| Language | English manuscripts | |
| Study period | From 2020 | |
| Geography | No restriction | |

"post-COVID", "COVID-19", etc). Similarly, the search term "Visuo*" will include "Visuo-spatial", "Visuospatial", "Visuo-constructive", etc.

The precise search strings generated from this technique were applied to each database and will be documented and reported in terms of the PRISMA-ScR framework[20] in the final review article.

### Primary search

A primary search conducted on 21 December 2023, resulted in the identification of 1409 articles. After an automated duplicate screening, 997 remained and were added to EndNote 21 to manage references and produce a bibliography for future screening. 925 records remained for screening after manual duplicate deletion, as depicted in figure 1.

### Screening and study selection

The review will be conducted in six stages: the problem is identified and the study design is being developed, the search strategy is established and implemented, literature

**Table 2** Conceptual search strategy and search terms

| Theme | Key concept | | Search terms/Strings in PubMed | Search terms/Strings in EMBASE |
|---|---|---|---|---|
| Long COVID | COVID | #1 | "COVID*"(TI) OR "SARS-CoV-2"(TI) | 'covid*':ti OR 'sars-cov-2':ti |
| | Infection | #2 | "COVID-19"(Mesh) OR "SARS-CoV-2"(Mesh) | 'coronavirus disease 2019'/exp |
| | Persistence | #3 | "long" OR "persistent" OR "chronic" OR "post-acute" OR "sequela*" | 'long' OR 'persistent' OR 'chronic' OR 'post-acute' OR 'sequela*' |
| | Long COVID | #4 | "Post-Acute COVID-19 Syndrome"(Mesh) | 'long covid'/exp |
| Cognition | Memory | #5 | "memor*"(TI) | 'memor*':ti |
| | Attention | #6 | "attention"(TI) | 'attention':ti |
| | Executive functioning | #7 | "executive function*"(TI) | 'executive function*' |
| | Vision | #8 | "visual" OR "visuo*"(TI) | 'visual':ti OR 'visuo*':ti |
| | Language | #9 | "language"(TI) OR "verbal"(TI) OR "fluency"(TI) OR "speech"(TI) | 'language':ti OR 'verbal':ti OR 'fluency':ti OR 'speech':ti |
| | Cognition | #10 | "cogniti*"(TI) | 'cogniti*':ti |
| | Mental state | #11 | "brain fog"(TI) OR "confusion"(TI) OR "orientation"(TI) OR "disorientation"(TI) | 'brain fog':ti OR 'confusion':ti OR 'orientation':ti OR 'disorientation':ti |
| Assessment | Paradigms | #12 | "neuropsycholog*"(TI) OR "neurocogniti*"(TI) | 'neuropsycholog*':ti OR 'neurocogniti*':ti |
| | Testbatteries | #13 | "MMSE" OR "MoCA" OR "CERAD" OR "TMT" OR "SDMT" OR "symbol digit modalities test" OR "Stroop" OR "N-BACK" OR "GO-NO*" OR "STMB* | 'mmse' OR 'moca' OR 'cerad' OR 'tmt' OR 'sdmt' OR 'symbol digit modalities test' OR 'stroop' OR 'n-back' OR 'go-no*' OR 'stmb*' |

The theme column specifies the specific topics of focus. The key concept column outlines the different notions covered by each corresponding theme.
The search terms/strings column provides: (1) the terms in the article either in any field, either in a specific field such as the title. (2) The exact standardised terms to index articles in the databases—they are assigned to articles to categorise their content in a consistent and organised manner. These standardised terms are known as MeSH terms (Medical Subject Headings) in MEDLINE/PubMed and EMTREE terms in EMBASE.

screening, data collection, data analysis, write-up and dissemination of findings.

Following the identification of articles matching the search strategy and duplicate removal, references in their totality will be documented and managed with the review software Rayyan.ai.[23] Articles will be systematically screened for eligibility in a stepwise manner by the reviewers. The number or studies after each step and exclusion decisions will be recorded, as illustrated in figure 1.

► Eligibility by title and abstract: two independent reviewers will be screening for titles and abstracts of the article, to identify eligibility for inclusion. The screening decisions will be compared between the two reviewers. A third independent reviewer will be consulted for any remaining discrepancies.

► Full text screening: the reviewer pair will assess the article selection for their full texts. Articles will be excluded if (1) data of interest is not reported, (2) patient population not eligible, (3) non-preferred study design, (4) non-preferred article type, or any other non-alignment with the eligibility criteria. Discrepancies will be solved as described in the previous section.

The final strategy combined successively all these search step as follows:

(((#1 OR #2) AND #3) OR #4) AND (#5 OR #6 OR #7 OR #8 OR #9 OR #10 OR #11 OR #12 OR #13).

**Data management (included and excluded papers)**

To maximise and document the objectivity of the screening process, the decisions and justifications for inclusion or

**Identification of Studies to be Included in the Scoping Review**

**Identification**

Search string and filters applied to databases

| PubMed | Embase |

Total records identified from databases
(n=1409)

Removal of duplicates:
Automated (n=412)
Manually (n=72)

Retrieved records to be screened
(n=925)

**Screening**

Screening records by titles and abstracts

Excluded records and reasoning

Total records sought for retrieval and for full text eligibility screening

Excluded records and reasoning

**Inclusion**

Total records included in the scoping review

**Figure 1** Flow diagram of the study selection methodology in concordance with the Reporting Items for Systematic Reviews and Meta-Analyses.[21]

exclusion will be documented in a digitalised spreadsheet table. Reference management, bibliography generation and write-up will be assisted by usage of EndNote 21, the screening will be conducted with Rayyan.ai [23] and a digital spread sheet will be used for charting the data.

### Data collection process and extraction (charting)

Rayyan.ai [23] and a digital spreadsheet will be used to extract and chart the data among the reviewer team members. All included papers will be reviewed by two

reviewers, each independently charting data for the following variables:

► Author(s).
► Year of publication.
► Title.
► DOI.
► Study objective.
► Study design (eg, cohort, multi-centre studies, cross-sectional, longitudinal, etc).

- ► Country of study.
- ► Setting context (eg, acute, primary healthcare, long term care, home or lab setting, etc).
- ► Sample size.
- ► Patient demographics (sex/gender (proportion), age, education (if present).
- ► Diagnostic group (if applicable; asymptomatic, mild, moderate, severe), comorbidity (if present, for example, pre-existing neurocognitive disorders), other characteristics related to long COVID (ie, impact of different variants of SARS-CoV-2, influence of COVID-19 vaccines or effects of reinfection).
- ► Symptom outcomes and time points or ranges (6, 12, 18, 24 or more, months).
- ► Outcome definition (including diagnostic criteria if available).
- ► Cognitive assessment methodology, measurement tools used, classification outcomes (with raw scores if included), rationale for including tests if stated.
- ► Statistical approach used.
- ► The authors' primary conclusions and, if available, explanations of the factors taken into consideration during the decision-making process.

The relevant items were adapted from the Cochrane Handbook for Systematic Reviews of Interventions.[24]

## Data analysis

To facilitate data synthesis, all the studies in review will be synthesised by being charted by the reviewers, to document the data of interest for reviewed studies and to visualise in the final report. In order to synthesise scoping evidence, both aggregative (summative) and configurative (organising) synthesis approaches will be used, to comprehensively map the field. The data analysis should both reveal gaps in the literature, but also magnitudes of evidence, as such a mixed approach of both evidence gap maps (EGM) and heat maps will be applied, along with other demographic visualisations as needed.

Specifically, EGM matrices will be populated to display presence or lack of evidence for cognitive outcomes, based on primary dimensions (duration, severity, tests used) and secondary dimensions (study design). This approach ensures plotting of heterogeneity in the field and gap identification, which aids at identifying research density. Heat map matrices will be used for various parameters depending on availability of data; for example, long COVID label characterisation (for duration, severity, study types), magnitude of symptoms reported per cognitive assessment and key impacting factors, with dimensions including SARS-CoV-2 impacting factors (influence of vaccines, variants, reinfections), comorbidities with respect to severity, duration and cognition reported, if sufficient data is present. Further layering of the matrix will be followed by group stratifications for identification of nuances.

This approach will support a quantifiable characterisation of the concepts, which aids at exploring the field and objectives, given the sufficiency of collected data.

Outcomes will be visualised in graphs and figures. The findings and figures will finally be disseminated in a narrative report, complementing the analysis by integrating findings across different methodologies and outcomes and compare them to existing models and theories.

## Quality assessment

In accordance with the guidelines developed by the Joanna Briggs Institute for scoping reviews,[25] no assessment of quality for included papers will be performed.

## Ethics and dissemination

The study does not directly involve human participants or personal data from potential participants, but only aggregated data from existing research. Therefore, no ethical approval is required. This protocol will enable for a scoping review whose results will be disseminated through its publication in a scientific journal, and within professional network.

**Contributors** Conceiving the idea of the scoping review and protocol: SM. Design of the protocol: SM, MP. Search strategy: SM, CD, MP. Draft of the manuscript: SM. Review and final approval of the manuscript: MP. ChatGPT was used for improving sentence structures.

**Funding** This work is supported by the Luxembourg National Research Fund (FNR), Project n°20210613.

**Competing interests** None declared.

**Patient and public involvement** Patients and/or the public were not involved in the design, or conduct, or reporting, or dissemination plans of this research.

**Patient consent for publication** Not applicable.

**Provenance and peer review** Not commissioned; externally peer reviewed.

**ORCID iDs**
Sara Monteiro http://orcid.org/0000-0001-9145-0063
Coralie Dessenne http://orcid.org/0000-0001-7723-0304
Magali Perquin http://orcid.org/0000-0002-0386-0212

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
