## [Reviewer comments · BMJ Open]

ARTICLE DETAILS

TITLE (PROVISIONAL)	Long COVID Cognitive Sequelae 6 Months Post-Infection and Beyond: A Scoping Review Protocol
AUTHORS	Monteiro, Sara; Dessenne, Coralie; Perquin, Magali

VERSION 1 – REVIEW

REVIEWER	Tebbutt, Scott The University of British Columbia
REVIEW RETURNED	05-Mar-2024

GENERAL COMMENTS	My Co This manuscript authored by Sara Monteiro and colleagues delineates a scoping review protocol aimed at investigating the cognitive sequelae associated with long COVID, with a specific focus on symptoms persisting six months or more post-infection. The following comments provide insights into areas for enhancement: Comparison with Similar Scoping Review Papers: The authors should explicitly reference existing review papers in the field and delineate the similarities and differences between their research plan and previously published works. For instance, references such as PMID: 34493506, PMID: 37351002 and PMID: 36579511. Inclusion of Alternate Labels for Long COVID: It would be beneficial for the authors to broaden their search strategy to encompass alternate labels for long COVID, including "Post-COVID Conditions," "long-haul COVID," "post-acute COVID-19," "long-term effects of COVID," and "chronic COVID." Integrating these terms into the search strategy (as detailed in Table 2) would facilitate a more comprehensive retrieval of pertinent literature, thereby enhancing the inclusivity of the review. Consideration of Key Factors: The protocol should address pivotal factors that could influence study outcomes but may not have been adequately considered. These factors encompass the impact of different variants of SARS-CoV-2, the influence of COVID-19 vaccines on long COVID symptoms, and the potential effects of reinfection or multiple reinfections. Furthermore, the protocol should recognize the role of patient comorbidities, such as neurocognitive disorders, in shaping cognitive outcomes associated with long COVID. Integrating these factors into the study design and analysis would bolster the comprehensiveness and relevance of the review findings.
--

	Anticipation of Potential Challenges in Data Synthesis: Although the paper outlines data extraction and synthesis methods, it is crucial to anticipate potential challenges in synthesizing heterogeneous data stemming from various study designs and populations. In the absence of a clear plan for addressing data heterogeneity and variability, the synthesis process may encounter obstacles, potentially compromising the robustness of the conclusions drawn from the review. In summary, while the outlined scoping review protocol addresses significant facets of long COVID cognitive sequelae, there are avenues for improvement in terms of acknowledging existing literature, broadening search terms, and considering key factors that may impact study outcomes. Incorporating these suggestions would augment the robustness and relevance of the review.
--	--

REVIEWER	SALMON, Dominique Assistance Publique - Hopitaux de Paris, Infectious and immunology Unit Hotel Dieu Hospital
REVIEW RETURNED	10-Mar-2024

GENERAL COMMENTS	The authors describe a protocol to better characterize cognitive impairment in PASC patients. This protocol is based on a systematic review of articles indexed in PUBMED focused in describing cognitive impairment in patients with PASC and using at least one validated and quantifiable cognitive scale. The authors plan that each of the articles identified by specific keywords be analyzed by two reviewers. Their objective is to better characterize cognitive disorders, correlate them with the severity of acute covid and describe the neurocognitive tests used to confirm the diagnosis of cognitive impairment. The research question is interesting. The work will allow to accurately describe the cognitive symptoms found in PASC patients, although I doubt that this description to go further than what we already know. The 2nd objective is to correlate the cognitive disorders with the severity of the acute COVID. This step needs to find articles that have addressed concomitantly these two questions but the authors do not present any data on the numbers of articles they have found that deal with these two questions at once. A third objective is to describe the contribution of quantitative tools to assess these cognitive disorders. But, outside the MoCa test that is known to be insufficiently precise for cognitive impairment in PASC, there are currently no validated quantitative tools. Other available tests, such as the neuropsychological assessment or the 18-FDG PET scan are not qualitative tests. Globally, the method in itself does not have an innovative or exceptional character that would justify a publication in BMJ open without any results. The readers of the BMJ open would certainly be interested to have a synthesis of the literature on the subject of cognitive disorders in PASC. I recommend to the authors to perform their study and then consider to submit it to the BMJ open
--

VERSION 1 – AUTHOR RESPONSE

RESPONSE TO REVIEWER 1: Dr. Scott Tebbutt, University of British Columbia

Dear Dr. Scott,

We greatly valued your kind and constructive feedback. Thank you for taking the time to provide your insights. Your comments have been extremely helpful and we have carefully revised our manuscript to address them, in order to improve the approach and rationale of our scoping study. Please find our responses and changes on a point-by-point basis below.

Comment 1: Comparison with Similar Scoping Review Papers: The authors should explicitly reference existing review papers in the field and delineate the similarities and differences between their research plan and previously published works. For instance, references such as PMID: 34493506, PMID: 37351002 and PMID: 36579511.

Response: Thank you for pointing out that we missed relevant articles, despite the literature review we initially conducted. Based on your suggestions, we added these three studies in the background section, to further elaborate on the type of scoping reviews already conducted, compared to the one we envisage. The following paragraph replaces the existing one, line [Pg 3, lines 22-24; 26-29], and the corresponding references (12, 13, 14) have been added to the bibliography section [Pg 14-15, lines 28-2]:

“Recent scoping reviews on COVID-19 reported cognitive symptoms and tests being used for screening up to six months maximum after the acute phase (9), as well as psychiatric conditions prior to and immediately after infection (12, 13). (...) However, a comparison between immediate and long-term effects has been reviewed, albeit on specific populations, such as patients with substance use disorders (14). Thus, although existing, to our knowledge, no study has addressed the specific and recurrent cognitive symptoms beyond 6 months after infection.”

Comment 2: Inclusion of Alternate Labels for Long COVID: It would be beneficial for the authors to broaden their search strategy to encompass alternate labels for long COVID, including "Post-COVID Conditions," "long-haul COVID," "post-acute COVID-19," "long-term effects of COVID," and "chronic COVID." Integrating these terms into the search strategy (as detailed in Table 2) would facilitate a more comprehensive retrieval of pertinent literature, thereby enhancing the inclusivity of the review.

Response: Thank you very much for bringing this forward, we realized that we may not have been clear enough in the manuscript. Indeed, it is very necessary not to be exclusive for alternate disease labels. In fact, we believe that our current search strategy already ensures the inclusion of alternate disease labels and we selected the current terms very carefully with assistance of our experienced scientific librarian to make sure our approach is correct. For instance, the search term “COVID*” will capture instances of the word "COVID," and it will also be inclusive of other labels including this term (e.g. long-COVID, post-COVID, COVID-19 etc.). Moreover, our key concept category “Persistence” includes single search terms that ensure the inclusion of alternate disease labels; e.g. “long”, “persistent”, “chronic”, “postacute” or “sequela” as described in Table 2. Therefore, the combination of these terms will lead to the inclusion of a majority of given conventional labels, which to our opinion match the current conventions in the field and the modification proposed here.

The Search strategy paragraph has now been modified to bring clarity with concrete examples [Pg 7, lines 9-12]:

“The search strategy was constructed together with a scientific librarian, to conduct a wideranging search within the selected databases, aiming at identification of an unbiased body of records. As outlined in Table 2, search terms from three themes (‘long COVID’, ‘cognition’ and ‘assessment’) will be conceptually arranged to be placed in comprehensive search strings for selection of relevant articles. Within a theme, search terms are arranged in conceptual clusters, which will be combined by syntax techniques such as Boolean operators (‘OR’, ‘AND’), search tags (MeSH, title only) and filters (‘most recent’, ‘humans’, ‘Eenglish language’). Additionally, word truncation () will be used to capture alternate word-endings. For example, the search term “COVID*” will capture instances of the word “COVID,” and also include other labels or terms containing “COVID” within them (e.g. “long-COVID”, “post-COVID”, “COVID-19” etc.). Similarly, the search term “Visuo*” will include “Visuo-spatial”, “Visuospatial”, “Visuoconstructive”, etc.”*

Additionally, a caption and details have been added to Table 2 [Pg 9, lines 1-9]:

“The Theme column specifies the specific topics of focus. The Key Concept column outlines the different notions covered by each corresponding Theme. The Search Terms/Strings column describes the exact standardized terms used by the National Library of Medicine (NLM) to index articles in the MEDLINE/PubMed database. These terms, known as MeSH terms (Medical Subject Headings), are assigned to articles to categorize their content in a consistent and organized manner.”

Comment 3: Consideration of Key Factors: The protocol should address pivotal factors that could influence study outcomes but may not have been adequately considered. These factors encompass the impact of different variants of SARS-CoV-2, the influence of COVID-19 vaccines on long COVID symptoms, and the potential effects of reinfection or multiple reinfections. Furthermore, the protocol should recognize the role of patient comorbidities, such as neurocognitive disorders, in shaping cognitive outcomes associated with long COVID. Integrating these factors into the study design and analysis would bolster the comprehensiveness and relevance of the review findings.

Response: This is a good point, as this type of analysis was planned but we might not have extensively discussed it here in the protocol. Initially, we decided not to overly focus on the impact of different variants, the influence of COVID vaccines, or the effects of reinfection, as the effects on cognition might be minimal or not adequately reported in studies. However based on your feedback and us mentioning this point in the background section, we have decided to include these in our data collection to make it clear, whether any data is reported in the context of cognitive changes.

These specifications have been added in “Diagnostic group” [Pg 11, lines 11-14]: *“Diagnostic group (if applicable; asymptomatic, mild, moderate, severe), comorbidity (if present e.g. pre-existing neurocognitive disorders), other characteristics related to long COVID (i.e. impact of different variants of SARS-CoV-2, influence of COVID-19 vaccines or effects of reinfection)”*

Further, we have extensively clarified this approach in our data analysis section [Pg 12, lines 1-18]:

“(…) In order to synthesize scoping evidence, both aggregative (summative) and configurative (organizing) synthesis approaches will be utilized, to comprehensively map the field. The data analysis should both reveal gaps in the literature, but also magnitudes of evidence, as such a mixed approach of both EGM and heat maps will be applied, along with other demographic visualizations as needed.

Specifically, EGM matrices will be populated to display presence or lack of evidence for cognitive outcomes, based on primary dimensions (duration, severity, tests used) and secondary dimensions (study design). This approach ensures plotting of heterogeneity in the field and gap identification. Heat map matrices will be used for various parameters depending on availability of data; e.g. long COVID label characterization (for duration, severity, study types), magnitude of symptoms reported per cognitive assessment and key impacting factors, with dimensions including SARS-CoV-2 impacting factors (influence of vaccines, variants, reinfections), comorbidities with respect to severity, duration and cognition reported, if sufficient data is present. Further layering of the matrix will be followed by group stratifications for identification of nuances. This approach will support a quantifiable characterization of the concepts, which aids at giving insights into the field and objectives, given the sufficiency of collected data. Outcomes will be visualized in graphs and figures. The findings and figures will finally be disseminated in a narrative report.”

Comment 4: Anticipation of Potential Challenges in Data Synthesis: Although the paper outlines data extraction and synthesis methods, it is crucial to anticipate potential challenges in synthesizing heterogeneous data stemming from various study designs and populations. In the absence of a clear plan for addressing data heterogeneity and variability, the synthesis process may encounter obstacles, potentially compromising the robustness of the conclusions drawn from the review.

Response: Thank you for warning us about these aspects. Anticipating potential challenges linked to heterogeneity and variability in detail may be premature until we have a comprehensive overview of data availability, which will become clearer as we proceed with the scoping review. However, your concerns and input have prompted us to reconsider and enhance our approach. We have decided to implement a new method involving evidence gap mapping techniques along with heat maps. This approach will provide visual and quantitative information on heterogeneity, enabling us to objectively assess the available knowledge and its quality within the field.

Anticipating potential challenges linked to heterogeneity and variability in detail may be premature until we have a comprehensive overview of data availability, which will become clearer as we proceed with the scoping review. However, your concerns and input have prompted us to reconsider and enhance our approach.

As heterogeneity can also be a strength for our scoping review (due to currently not too many studies focusing on cognitive long-term outcomes), we aim not to be too restrictive here in our approach. As one of our objectives is to elucidate gaps in the literature, we have decided to implement an evidence gap mapping technique along with our heat maps, to provide visual and quantitative information on heterogeneity, which will also help to objectively reflect on the available knowledge and quality within the field. Further, depending on data availability we will conduct further layering's of our data from more general concepts such as memory, to more nuanced properties.

We have elaborated extensively in the data analysis section **[Pg 12, lines 1-18]:**

“(…) In order to synthesize scoping evidence, both aggregative (summative) and configurative (organizing) synthesis approaches will be utilized, to comprehensively map the field. The data analysis should both reveal gaps in the literature, but also magnitudes of evidence, as such a mixed approach of both EGM and heat maps will be applied, along with other demographic visualizations as needed.

Specifically, EGM matrices will be populated to display presence or lack of evidence for cognitive outcomes, based on primary dimensions (duration, severity, tests used) and secondary dimensions (study design). This approach ensures plotting of heterogeneity in the field and gap identification. Heat map matrices will be used for various parameters depending on availability of data; e.g. long COVID label characterization (for duration, severity, study types), magnitude of symptoms reported

per cognitive assessment and key impacting factors, with dimensions including SARS-CoV-2 impacting factors (influence of vaccines, variants, reinfections), comorbidities with respect to severity, duration and cognition reported, if sufficient data is present. Further layering of the matrix will be followed by group stratifications for identification of nuances. (...)”

Comment 5: In summary, while the outlined scoping review protocol addresses significant facets of long COVID cognitive sequelae, there are avenues for improvement in terms of acknowledging existing literature, broadening search terms, and considering key factors that may impact study outcomes. Incorporating these suggestions would augment the robustness and relevance of the review.

Response: We hope to have satisfied your concerns and remain open for further suggestions. We reiterate our gratitude for your suggestions for improvement. Thanks to them, we have:

- Enriched the manuscript with additional existing studies (see response to comment 1),
- Clarified the approach we intend to use with broad search terms (see response to comment 2),
- Incorporated key factors relevant for the study outcomes into the analysis (see response to comment 3).

RESPONSE TO REVIEWER 2: Dr. Dominique SALMON, Assistance Publique - Hopitaux de Paris

Dear Dr. Salmon,

We thank you for your critical appraisal of our scoping review protocol, and the time you have invested into it. Your comments have helped us reflect on our rationale and make adjustments accordingly. Please find our responses in blue for each comment below.

Comment 1: The research question is interesting. The work will allow to accurately describe the cognitive symptoms found in PASC patients, although I doubt that this description to go further than what we already know.

Response: Thank you for your acknowledgement of our research question and of our review. One of our objectives is indeed to reflect on the current gaps in the literature to provide an overview for researchers and clinicians alike to understand where more work will be needed. Further, similar scoping studies indeed already exist in the context of symptomatology present up to 6 months, while no current review addresses cognitive symptoms beyond. We therefore believe that our approach is legitimate and novel in the context of existing studies and kindly ask you to reconsider.* (*please also see reply to comment 2*)

Medical studies currently target psychometric approaches and cognitive symptoms, many times, “less well”, which is one of the points we aim to appraise critically with our study. We further would like to ensure the credibility of our approach and study in the open science context, by making our specific approach available before our final article, as we try to understand not only cognition from a multidimensional perspective, but also everything related to it. This will provide a very granular

understanding of cognition in this context and might find other applications for diseases or characteristics in the design of scoping studies, since currently many studies that are conducted outside of the conventional field of psychology indeed treat cognition as an umbrella term, without highlighting nuances, which hereby we try to standardize more for future studies.

Comment 2: The 2nd objective is to correlate the cognitive disorders with the severity of the acute COVID. This step needs to find articles that have addressed concomitantly these two questions but the authors do not present any data on the numbers of articles they have found that deal with these two questions at once.

Response: Indeed, we did not originally present this in the current protocol as we have not currently commenced the screening of papers, but we will ensure to be comprehensive about the amount of papers we have found addressing each specific objective during the analysis and write-up phase. In order to be more precise about our approach here, we have elaborated on an **evidence gap map approach** that we will implement, which will also help with assessing the number of articles for various primary and secondary variables, including those addressing both cognition and severity.

Your comment helped us to make a decision here on the necessity of including this approach, as indeed the data will lead to clearer visualizations for questions of amount of articles and article types, and will help us identify what is currently known and what should be further investigated (**connecting back to your comment 1***). In order to highlight this, we have included this into under the section “Data analysis” [Pg 12, lines 1-18]:

“(...) In order to synthesize scoping evidence, both aggregative (summative) and configurative (organizing) synthesis approaches will be utilized, to comprehensively map the field. The data analysis should both reveal gaps in the literature, but also magnitudes of evidence, as such a mixed approach of both EGM and heat maps will be applied, along with other demographic visualizations as needed.

Specifically, EGM matrices will be populated to display presence or lack of evidence for cognitive outcomes, based on primary dimensions (duration, severity, tests used) and secondary dimensions (study design). This approach ensures plotting of heterogeneity in the field and gap identification. Heat map matrices will be used for various parameters depending on availability of data; e.g. long COVID label characterization (for duration, severity, study types), magnitude of symptoms reported per cognitive assessment and key impacting factors, with dimensions including SARS-CoV-2 impacting factors (influence of vaccines, variants, reinfections), comorbidities with respect to severity, duration and cognition reported, if sufficient data is present. Further layering of the matrix will be followed by group stratifications for identification of nuances. (...)”

Comment 3: A third objective is to describe the contribution of quantitative tools to assess these cognitive disorders. But, outside the MoCa test that is known to be insufficiently precise for cognitive impairment in PASC, there are currently no validated quantitative tools. Other available tests, such as the neuropsychological assessment or the 18-FDG PET scan are not qualitative tests.

Response: We fully agree with the first statement, which motivated us to also assess which psychometric tests are used, to make inferences about cognition. We believe that this is where we currently see much room for improvement in the field. Currently, there is no standard protocol in the context of long COVID. Therefore, we think it is highly relevant to scope: 1) the type of cognitive long-term complications, 2) the methodologies currently being used for such claims, 3) other potential influences on current estimates about cognition (as a gap is still remaining in the scientific literature and in the clinical consensus. Nicotra et al. (2023) have made an attempt at describing a potential cognitive screening protocol, however the provided reasoning is improvable, which we aim to develop

by adding the very long term complications and more overall nuance in influencing factors. This protocol, as mentioned in our comment above, would thus also help provide a more comprehensive mapping of the field, highlighting key aspects from the neuropsychological perspective on current studies.

Comment 4: Globally, the method in itself does not have an innovative or exceptional character that would justify a publication in BMJ open without any results.

Response: We hope we have provided enough context to demonstrate the relevance of our scoping review protocol. It aims to address methodological shortcomings in scoping methodology, and meet the clinical and scientific necessity to fill the current research gap. By additionally plotting evidence gap maps, we aimed to address your concerns about the lack of clarity regarding amount of evidence per variable, the lack of innovation and reflection on evidence available for each neuropsychological assessment. We thank you for your concerns and your feedback and remain at your disposal for further comments or questions.

Comment 5: The readers of the BMJ open would certainly be interested to have a synthesis of the literature on the subject of cognitive disorders in PASC. I recommend to the authors to perform their study and then consider to submit it to the BMJ open

Response:

We appreciate your interest in our scoping review project, and your suggestion to consider submitting the study itself to BMJ Open. While we understand your viewpoint, we believe that publishing the protocol of a complex study is valuable as it provides transparency and ensures adherence to rigorous methodological standards. Nonetheless, we will consider your recommendation carefully as we move forward with our research. Thank you for your thoughtful feedback, through which we have made the following adjustments:

VERSION 2 – REVIEW

REVIEWER	Tebbutt, Scott The University of British Columbia
REVIEW RETURNED	22-Apr-2024

GENERAL COMMENTS	Assistance with this review was from Dr. Chengliang Yang (UBC). The revised scoping review protocol on long COVID cognitive sequelae six months post-infection and beyond is comprehensive and methodologically rigorous. Overall, this protocol is well-prepared, demonstrating a clear, structured approach to understanding a complex and emerging area of research. To further enhance its rigour and utility, consider addressing the following areas before acceptance for publication: Quality Assessment and Bias of Included Studies: Although scoping reviews typically do not include a risk of bias assessment, incorporating a quality appraisal could be beneficial. This could involve summarizing the strength of the evidence in a table format, showing each included study alongside several quality criteria relevant to that study type. Such an assessment can guide future systematic reviews or primary studies by highlighting areas in need of high-quality research. Data Synthesis and Interpretation: The use of Evidence Gap Maps (EGM) and heat maps for data synthesis is commendable. However, the protocol would benefit from additional details on how these tools will be employed to address heterogeneity in study designs, populations, and outcomes. Further clarifications on how the findings will be interpreted in the context of existing theories and models of long COVID could also strengthen the review. Minor Issues: Page 8, line 4: This line contains an incomplete sentence that needs to be revised for clarity. Page 9, line 7: Replace the comma with a period to correct the sentence structure.
---

VERSION 2 – AUTHOR RESPONSE

RESPONSE TO EDITORS, Dr. Thomas Phillips, Dr. Chengliang Yang

Comment 1:

Quality Assessment and Bias of Included Studies: Although scoping reviews typically do not include a risk of bias assessment, incorporating a quality appraisal could be beneficial. This could involve summarizing the strength of the evidence in a table format, showing each included study alongside several quality criteria relevant to that study type. Such an assessment can guide future systematic reviews or primary studies by highlighting areas in need of high-quality research.

To go along with your point, we believe that the EGM approach adds a preliminary layer of evidence strength that can incentivize further systematic reviews. Although the suggestion is commendable for

systematic reviews and meta-analyses, for our present design it is outside of our methodology and rationale. Furthermore, we were aiming to implement the JBI guideline criteria by Peters et al. (2015) for scoping reviews, according to which the critical appraisal of study quality is not typical: “*Another distinction between scoping reviews and systematic reviews is that unlike a systematic review, scoping reviews are designed to provide an overview of the existing evidence base regardless of quality. Hence, a formal assessment of methodological quality of the included studies is generally not performed.*”

The goal of our scoping review is to map the literature and identify gaps compared to addressing strength of evidence. Nevertheless, the quality of cognitive assessments used and the variability in studies, will be critically appraised in our narrative synthesis. As such, we are concerned about adding a supplementary table showing several quality criteria for each included study, as it may appear redundant here.

Further quality insurance is provided by clear inclusion and exclusion criteria (as such, exclusion of low-quality studies). A rigorous collection of data will be performed, which will be further synthesised and assessed for gaps narratively.

Peters MDJ, Godfrey CM, Khalil H, McInerney P, Parker D, Soares CB. Guidance for conducting systematic scoping reviews. *Int J Evid Based Healthc.* 2015 Sep;13(3):141–6.

Comment 2:

Data Synthesis and Interpretation: The use of Evidence Gap Maps (EGM) and heat maps for data synthesis is commendable. However, the protocol would benefit from additional details on how these tools will be employed to address heterogeneity in study designs, populations, and outcomes. Further clarifications on how the findings will be interpreted in the context of existing theories and models of long COVID could also strengthen the review.

Thank you for this valuable feedback. Due to heterogeneity, a narrative synthesis seems to be the best approach, combined with EGMs to assess the quantity of evidence and heat maps to visualize the objectives. We would also like to highlight again, that the overarching theme of the study is neurocognitive long-term changes, with a scoping review focusing on identifying key characteristics thereof, gaps in research and evidence types, serving as a groundwork for more detailed approaches.

We have changed the following sentences on page 12 to nevertheless clarify:

- *Lines 8-9: This approach ensures plotting of heterogeneity in the field and gap identification, **which aids at identifying research density.***
- *Lines 19-20: The findings and figures will finally be disseminated in a narrative report, **complementing the analysis by integrating findings across different methodologies and outcomes and compare them to existing models and theories.***

We hereby elaborate on our reasoning:

However, the protocol would benefit from additional details on how these tools will be employed to address heterogeneity in study designs, populations, and outcomes.

Line 6-9: “Specifically, EGM matrices will be populated to display presence or lack of evidence for cognitive outcomes, based on primary dimensions (duration, severity, tests used) and secondary dimensions (study design). This approach ensures plotting of heterogeneity in the field and gap identification, which aids at identifying research density.”

Primary: Cognition, Duration		Secondary: Study type		
 ■ Primary Studies ■ Commentaries ■ Reviews 	Cognition			
	Duration	Memory	Attention	X
6	 1 17 23 			
12	 0 2 1 			
18				

We here lay out that the primary dimensions represent the variables on the axes of the table, while secondary dimensions are e.g. bar graphs in the fields, for each study type representing a frequency count. The same can be visualized for other types of data, however as we are not sure about data quality at present, we would not like to overcommit at the present stage. An example layout (which as mentioned, can be layered further depending on data availability):

Line 9-15: “Heat map matrices will be used for various parameters depending on availability of data; e.g. long COVID label characterization (for duration, severity, study types), magnitude of symptoms reported per cognitive assessment and key impacting factors, with dimensions including SARS-CoV-2 impacting factors (influence of vaccines, variants, reinfections), comorbidities with respect to severity, duration and cognition reported, if sufficient data is present. Further layering of the matrix will be followed by group stratifications for identification of nuances.”

Heat maps will be used to collect data on the concepts such as label, cognitive skills assessed, psychometric tests used, etc.

It is important to note that as this is a protocol for a scoping review aimed primarily at mapping cognitive long-term changes in long COVID, we are cautious about overcommitting to specific analytical frameworks at this stage, given potential limitations in data availability.

Further clarifications on how the findings will be interpreted in the context of existing theories and models of long COVID could also strengthen the review.

We appreciate your suggestion about the comparison to existing models, and tried to elaborate accordingly. We currently view the natural heterogeneity of the field as part of our mapping approach. We would like to compare current models and labels of long COVID theoretical models, and currently do not already choose one prior to our analysis; instead we aim to provide insights into currently available evidence for each of these models (e.g. comparing various theories and reported duration, study types, severity...; depending on availability of data to report). This can entail the critical appraisal of existing frameworks and theories or the attempt to update existing models and approaches as a consequence of our findings.

Minor comments:

The changes were made accordingly. On page 8, the figure shifted above the already present text, we have put it back in its position. The dot was added on page 9 line 5 for correct punctuation.